# The Recurring Loss of ORF8 Secretion in Dominant SARS-CoV-2 Variants

**DOI:** 10.3390/ijms26125778

**Published:** 2025-06-16

**Authors:** Joy-Yan Lam, Kin-Hang Kok

**Affiliations:** 1Department of Microbiology, Li Ka Shing Faculty of Medicine, The University of Hong Kong, Hong Kong SAR, China; 2Centre for Virology, Vaccinology and Therapeutics, Hong Kong Science and Technology Park, Hong Kong SAR, China; 3Pandemic Research Alliance Unit, The University of Hong Kong, Hong Kong SAR, China; 4State Key Laboratory for Emerging Infectious Diseases, The University of Hong Kong, Hong Kong SAR, China; 5AIDS Institute, Li Ka Shing Faculty of Medicine, The University of Hong Kong, Hong Kong SAR, China

**Keywords:** COVID-19, SARS-CoV-2, ORF8, virus evolution

## Abstract

The SARS-CoV-2 ORF8 protein is a unique accessory viral protein among human coronaviruses, characterized by recurrent deletions and mutations with functional consequences. In this short report, we demonstrate that several dominant SARS-CoV-2 strains, despite encoding ORF8, fail to secrete the protein, revealing a recurring pattern of ORF8 functional impairment that cannot be detected by sequence analysis alone. In agreement with other studies, several high-frequency mutations were identified using the Nextstrain/augur pipeline, including G8Stop, Q27Stop, D119-/F120- double deletions, and nucleotide substitution C27889U, which occurred in XBB.1.5, Alpha, Delta, and BA.5.2 variants, respectively. Notably, the D119-/F120- deletions and C27889U substitution do not introduce premature stop codons, yet ORF8 secretion was lost in Delta and BA.5.2 virus-infected cultures. This indicates that the extracellular ORF8 function is impaired in these variants, resulting in ORF8 deficiency. Our findings highlight that the impairment of ORF8 secretion arises not only from premature stop codons but also from other mutations. Therefore, the functional validation of ORF8 secretion and activity is essential following sequence analysis to accurately assess ORF8’s role in SARS-CoV-2 infection.

## 1. Introduction

Monitoring SARS-CoV-2 evolution remains an important task even after the global pandemic. Similar to other infectious viruses, such as influenza, SARS-CoV-2 continues to evolve, evade pre-existing or vaccine-induced immunity, and cause a constant health burden worldwide. Understanding how SARS-CoV-2 mutates helps to gain knowledge on its virology and thus better prepare for the emergence of future variants.

SARS-CoV-2 ORF8 protein is a cryptic protein and is more closely related to bat-related coronaviruses [1]. ORF8 protein demonstrates multifunctional roles and is involved in various processes, such as impairing host antigen presentation [2,3], inducing inflammation and cytokine storm [4,5], acting as viral mimicry [6,7], as well as binding to immune cells such as dendritic cells and monocytes [8,9]. Previous studies from our group and others confirmed the presence of the extracellular ORF8 protein, leading to the fact that ORF8 is the first secreted viral protein among human coronaviruses [10]. We also found that convalescent patient sera contain significant levels of anti-ORF8 antibodies, indicating that ORF8 is highly immunogenic. This reflects that ORF8 plays a role in both viral pathologies and host immunity. But whether it is essential or dispensable to the virus’s fitness remains to be answered, especially when there is growing evidence that ORF8 is repeatedly lost in circulating SARS-CoV-2 strains [11].

In this short report, we present in vitro evidence demonstrating that, despite retaining intact ORF8 sequences, certain dominant SARS-CoV-2 variants have lost the ability to secrete ORF8, indicating impaired extracellular ORF8 function. Analysis of supernatants from virus-infected cells revealed impaired ORF8 secretion in the Delta and BA.5.2 variants, both of which encode ORF8 but harbor distinct mutations. Using the Nextstrain pipeline, we identified four high-frequency mutations consistent with previous studies: Q27Stop, G8Stop, the D119-/F120- double deletion, and the nucleotide substitution C27889U, which are found in the XBB.1.5, Alpha, Delta, and BA.5.2 variants, respectively. Notably, these findings suggest that ORF8 function is compromised not only by premature stop codons but also by other mutations. Our study provides molecular evidence visualizing the recurrent loss of ORF8 secretion throughout SARS-CoV-2 evolution and underscores the importance of validating ORF8 secretion in future research.

## 2. Results

We experimentally determined the secretion of ORF8 in the major SARS-CoV-2 variants (Figure 1A). VeroE6-TMPRSS2 cells were infected with respective SARS-CoV-2 variants at MOI 1.0. At 24 h, culture supernatant was harvested and probed for secreted ORF8 expression using Western blot. The presence of secreted ORF8 was observed for Ancestral, Beta, Omicron BA.1/BA.2, and JN.1. As expected, extracellular ORF8 was not detected in the Alpha variant (Q27Stop), as well as XBB.1.5 and EG.1.5 (G8Stop), where premature stop codons were present. Interestingly, the secretion of ORF8 was not detected in Delta (D119-/F120-), BA.5.2, and BQ.1.1 (C27889U).

To put our observation in the context of ORF8 evolution, representative SARS-CoV-2 genome sequences were selected and analyzed based on the data curated at Nextstrain.org, accessed on 11 November 2024 (Appendix A). Raw sequence files were obtained from GISAID. A total of 2033 sequences were obtained, which include the ancestral virus (WH01, EPI_ISL_406798), Alpha (109), Beta (38), Delta (481), BA.1 (132), BA.2.75 (58), BA.5 (511), BQ.1 (59), XBB (61), XBB.1.5 (141), EG.1.5 (40), HK.3 (63), BA.2.86 (14), JN.1 (286), KP.3 (6), XDV.1 (26), and XEC (7) variants. Together with the root sequences from Nextstrain SARS-CoV-2 workflow (204), we performed phylogenetic analysis (total 2237 sequences) using the Nextstrain pipeline, involving Augur tools and Auspice visualization. Similar to other studies, distinctive mutations were observed (Figure 1B). First, the Alpha variant carried Q27Stop mutation and circulated from the end of 2020 to mid-2021. Second, D119-/F120- double deletion, which contributes to the loss of salt bridges for dimerization [12], was observed in the Delta variant and circulated from 2021 to 2022. Third, the nucleotide substitution C27889U was observed in the BA.5.2 variant, which is a substitution that occurred in the TRS-B region upstream of the ORF8 coding sequence. BA.5.2 dominated from 2022 to 2023. Fourth, G8Stop mutation was observed in the XBB.1.5 and its descendants, EG.1.5 and HK.3, which co-circulated from 2023 to 2024.

Taken together, the evolution of secretory ORF8 was visualized (Figure 1C). SARS-CoV-2 began with ORF8 secretion (Ancestral), followed by an absence of secretion for 1 year (2021–2022, Alpha and Delta), and then was restored during the following ~6 months (2022–2022 Mid, BA.1 and BA.2). The secretion was lost again in the subsequent 1.5 years (2022 Mid-2024, BA.5 and XBB.1.5), only to be restored afterwards (2024–2025, JN.1).

## 3. Discussion

In this short study, we examine SARS-CoV-2 evolution through the lens of ORF8 protein secretion and provide experimental evidence that the Delta and BA.5.2 variants exhibit impaired ORF8 secretion. Similar to other peer-reviewed studies, the recurring disruption of ORF8 during SARS-CoV-2 evolution [11] and various mutations were reported [13,14,15]. However, there is less focus on experimental evidence regarding protein secretion, which is a characteristic property of ORF8.

The function of the ORF8 protein is thought to be related to antigen presentation, cytokine storms, and inflammation. Studies of its function have presented various mechanisms, demonstrating the multifunctional role of ORF8, but there has yet to be a consensus across studies. There are two major structural properties of the ORF8 protein: secretion to the extracellular space and dimerization. Secretion plays a crucial role in its host-disrupting function, such as binding to innate cells, inducing inflammation, and viral mimicry to IL17RA [7,8,9]. However, it is important to understand whether ORF8 is being secreted during infection, even when an intact coding sequence is observed, and as many studies utilize plasmid expression to investigate ORF8 function. Our findings indicated that ORF8 secretion was lost in Delta and BA.5.2 variants, suggesting that mutations other than the introduction of premature stop codons can compromise extracellular ORF8 function. Hence, our protein analysis data connects the gap between sequence analysis and live virus infection and provides insights into the impact of ORF8 secretion in different variants.

The retrospective analysis of ORF8 evolution (Figure 1C) reveals a dynamic pattern: the ancestral SARS-CoV-2 virus was initially supplanted by variants characterized by the loss of ORF8 secretion, notably Alpha and Delta. These were subsequently replaced by Omicron sub-lineages BA.1 and BA.2, which restored the secretion. This phase was followed by the emergence of ORF8-deficient variants BA.5.2 and XBB.1.5. Most recently, as of 2025, full-length ORF8 variants such as JN.1 and its descendants XEC have risen to prominence. This cyclical pattern underscores the recurrent gain and loss of extracellular ORF8 functionality throughout SARS-CoV-2 evolution, reflecting complex selective pressures acting on this accessory protein. The loss of secretion and premature stop codons represent ORF8 deficiency, and it is not transient; variants dominated for around 1 year, reflecting that ORF8 is dispensable to some extent, but the virus “rescues” it eventually. This repeating pattern of ORF8 deletion has been recognized previously based on premature stop codons and deletions in sequences. Nevertheless, our data here adds insight to this dynamic pattern, as some variants without premature stop codons also have impaired ORF8 function. This may also suggest that ORF8 has the role of a “burner” gene in the virus genome such that SARS-CoV-2 can use to compensate for other beneficial mutations in its genome to ensure virion stability and successful virion packaging, as several studies have successfully inserted exogenous sequences in the ORF8 reading frame to generate recombinant viruses [16,17]. However, it is important to note that the extracellular function of ORF8 is still somewhat ambiguous. While some studies present evidence on cytokine disruption by extracellular ORF8 [7,9], some studies found that secreted ORF8 does not significantly affect the cytokine profile [18]. Whether the loss of secreted ORF8 only contributes to disease severity warrants further investigation.

In summary, this short study highlights the impaired ORF8 secretion in dominant SARS-CoV-2 strains, including both the premature stop codons that abolish expression and other mutations, which demonstrate ORF8 deficiency. This underscores the importance of validating ORF8 characteristics when studying its function. While the impact of recurring ORF8 deficiency can be difficult to discern, as viral fitness is still largely determined by the competency of the spike protein, it will be essential to investigate the long-term role of ORF8 in the SARS-CoV-2 genome. Observing the consistency of recurring patterns and elucidating the true role of ORF8 will provide crucial insights into its significance in viral biology and evolution.

## 4. Materials and Methods

### 4.1. Sequence Analysis

In total, 2037 sequences were acquired from GISAID and fed into Nextstrain pipeline with a SARS-CoV-2 workflow [19,20]. Frequency panels were generated by auspice 2.0 running on the local server. Sequences from each variant were aligned using MAFFT. Consensus ORF8 protein sequences of each variant were then aligned for phylogenetic tree analysis using MEGA 11 (v11.0.13) and Treeviewer (v2.2.0) software.

### 4.2. Protein Analysis in Infected Cells

VeroE6-TMPRSS2 cells were obtained from the Japanese Collection of Research Bioresources (JCRB) cell bank and cultured in DMEM medium supplemented with 10% fetal bovine serum. Virus stocks were prepared and plaque-purified in VeroE6-TMPRSS2 cells. All virus culturing and infection experiments were performed in the BSL-3 laboratory at Queen Mary Hospital. Cells were inoculated with respective virus strains at MOI 0.1 for 1 h. The inoculum was then removed; cells were washed with PBS and were replenished with fresh DMEM/1%FBS. Culture supernatants were harvested at 24 h post-infection and centrifuged to remove cell debris. Cells were lysed by RIPA buffer on ice. All samples were heat-denatured in the presence of a reducing agent (5% 2-mercaptoethanol) and separated by denaturing SDS-PAGE. The presence of proteins was visualized by Western blot. ORF8 was detected by anti-ORF8 antibody (Invitrogen, Waltham, MA, USA, PA5-143463), while nucleocapsid protein was detected by anti-nucleocapsid antibody (SinoBiological, Beijing, China, 40143-R001). β-actin was detected by anti-actin antibody (Sigma, St. Louis, MO, USA, A5316). Detection was performed using chemiluminescence with HRP-conjugated secondary antibody (Amersham, Chalfont St. Giles, UK, NA934) and WesternBright ECL (Advansta, San Jose, CA, USA), and visualized using Sapphire FL Biomolecular Imager (Azure Biosystems, Dublin, CA, USA).

## Figures and Tables

**Figure 1 ijms-26-05778-f001:**
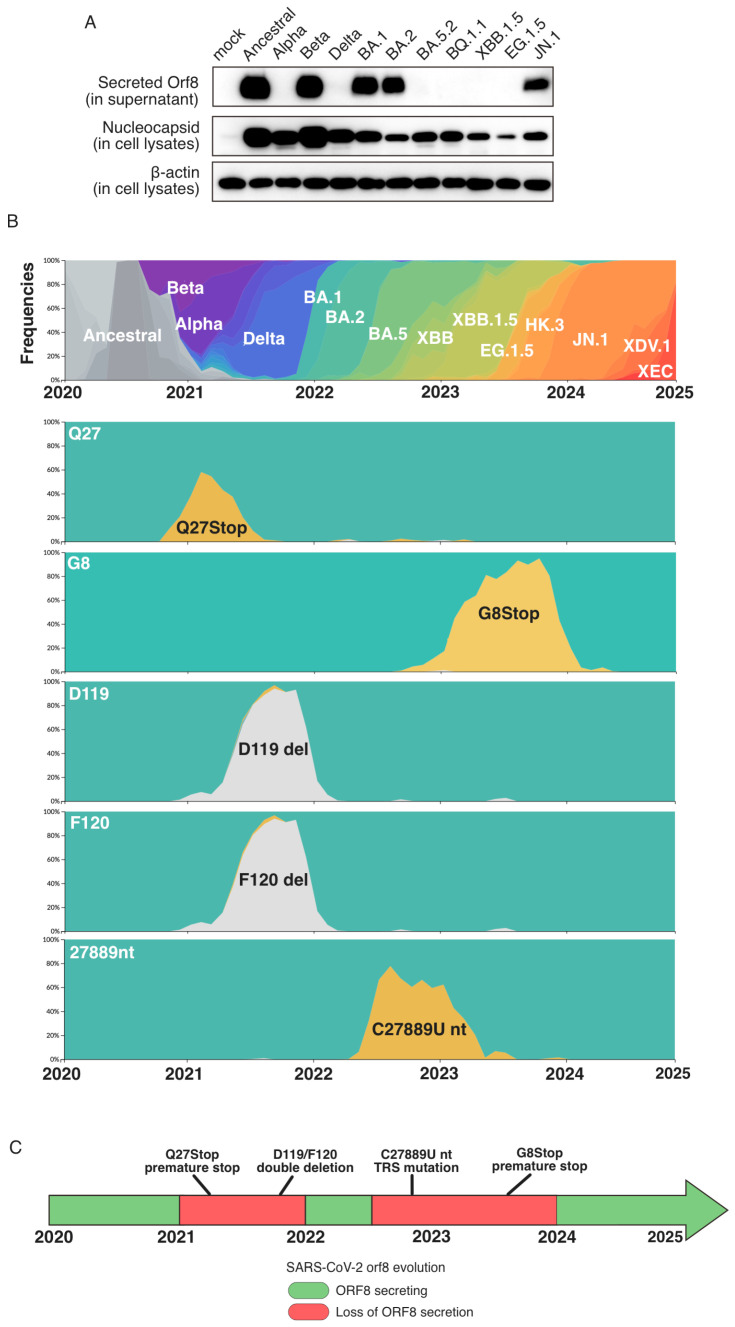
Analysis of ORF8 protein secretion in dominant SARS2 strains. (**A**) Western blot analysis of infected VeroE6-TMPRSS2 cells. Samples were harvested 24 h post-infection. ORF8 was detected in the culture supernatant, while nucleocapsid protein was detected in cell lysates. (**B**) Nextstrain/Augur pipeline analysis of SARS2 genomic sequences from 2020 January to 2024 November. Frequency tables were presented, labeled with respective mutation. (**C**) Illustration of the evolution of ORF8 secretion. The bar illustration represents ORF8 secreting (green) or loss of ORF8 secretion (red).

## Data Availability

Data is contained within the article or Appendix A.

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
