# Peer review of "The Recurring Loss of ORF8 Secretion in Dominant SARS-CoV-2 Variants"

_ijms, 2025, doi:10.3390/ijms26125778_

Round 1
Reviewer 1 Report (New Reviewer)
Comments and Suggestions for Authors
Review report on the Brief Report entitled “The recurring loss of ORF8 secretion in dominant SARS-CoV-2 variants”
The authors investigated the secretion of ORF8 in SARS2 strains. This brief report highlights the importance to incorporate methods to detect secreted ORF8 in experiments and not make assumptions based on ‘intact ORF8 sequences’ alone.
The authors demonstrated with multiple SARS2 genomic sequences from 2020 to 2024/25 in Figure 1, the almost yearly cycle of loss and restoration of ORF8 secretion. To a certain extent, in my opinion, this cycle of loss/restoration of extracellular ORF8 gives the appearance that this protein potentially contributes to overall viral fitness* (possibly transmission fitness).
*Viral fitness used to be and still is for the most part, mainly determined by replicative fitness. However, research in this field has shown that various factors are involved in overall viral fitness, including transmission fitness.
The focus and outcome of this investigation was very interesting. Hopefully, the authors can provide definitive answers about this ‘seeming pattern’ of loss/restoration of extracellular ORF8 in future studies.
Author Response
Thank you very much for reviewing and commenting on our brief report. We also believe that the recurring pattern of orf8, especially for the secretion properties, should have some impact on the virus’s fitness. Possibly, it can only be revealed in the long term. Hopefully, we and the other researchers can continue to uncover the true role of ORF8. Based on other reviewers’ comments, we have loading control for the Western Blot.
Reviewer 2 Report (New Reviewer)
Comments and Suggestions for Authors
The manuscript by Lam and colleagues described the dynamics of ORF8 secretion and explored the possible mutants underlying the loss of secretion. The manuscript provided evidence showing the loss of secretory form of ORF8 in major Sars-cov2 variants.
Overall, this brief report is well-organized and provided molecular evidence that ORF secretion was affected by mutations causing premature stop and mutations in other key sites. The data was consistent with genomics data and retrospective dynamic secretion pattern was discussed. While ORF8 has been reported to has signal peptide and be secreted from infected cells, this study adds minimal to literature.
Author Response
Thank you for reviewing and giving comments on our brief report. We fully understand that the repeating pattern of ORF8 knockout and ORF8 secretion has been described in other studies. Here in our short study, we focus on the changes in ORF8 secretion among the variants. We are interested in this novel viral protein in human coronaviruses and will be conducting more characterization. Based on other reviewers’ comments, we have loading control for the Western Blot. Hopefully, we can provide more findings and report in a longer format in the future.
Reviewer 3 Report (New Reviewer)
Comments and Suggestions for Authors
The authors aimed to investigate the loss of ORF8 secretion in SARS-CoV-2 variants to discuss how ORF8 might relate to the fitness of the virus. However, this works shows limited scientific rigor, compromising the validness of its conclusions. I would not recommend this manuscript to be published. My specific concerns are followed:
Figure 1A lacks a loading control, such as actin. Also, did the authors repeat the Western blot to show statistical consistency? Most importantly, is it possible that the ORF8 antibody used could not recognize certain mutated ORF8? How could the authors exclude this possibility? Could the authors use the same antibody to detect the intracellular ORF8?
The mutations detected are not carried through time. Is it accurate to conclude these mutations are the consequences of ‘evolution’?
Author Response
Please refer to the attached reply document. Thank you.

Round 2
Reviewer 3 Report (New Reviewer)
Comments and Suggestions for Authors
Thank the authors for their detailed elaborations. I believe this work is publishable as a brief report.
This manuscript is a resubmission of an earlier submission. The following is a list of the peer review reports and author responses from that submission.
Round 1
Reviewer 1 Report
Comments and Suggestions for Authors
In this brief report, the authors investigate the mutational trends in the ORF8 gene of SARS-CoV-2 over time. They note that there are many loss-of-function mutations in ORF8, leading to an "ORF8 deficiency".
While the work is relatively sound (albeit lacking depth), it lacks any novelty. The exact topic, i.e. the loss/impaired function of ORF8 in SARS-CoV-2 over time, has been addressed in several to many studies before. Either the authors were unaware of the existing literature, or chose not to cite previous studies investigating this topic to give the illusion of increased novelty. Some examples of relevant studies (from as far back as 2021):
- Arduini A, Laprise F, Liang C. SARS-CoV-2 ORF8: A Rapidly Evolving Immune and Viral Modulator in COVID-19. Viruses. 2023 Mar 29;15(4):871. doi: 10.3390/v15040871. PMID: 37112851; PMCID: PMC10141009.
- Hisner R, Gueli F, Peacock T. Repeated loss of ORF8 expression in circulating SARS-CoV-2 lineages - SARS-CoV-2 coronavirus. Virological. 2023
- Wagner C, Kistler KE, Perchetti GA, Baker N, Frisbie LA, Torres LM, Aragona F, Yun C, Figgins M, Greninger AL, Cox A, Oltean HN, Roychoudhury P, Bedford T. Positive selection underlies repeated knockout of ORF8 in SARS-CoV-2 evolution. Nat Commun. 2024 Apr 13;15(1):3207. doi: 10.1038/s41467-024-47599-5. PMID: 38615031; PMCID: PMC11016114.
- Zinzula L. Lost in deletion: The enigmatic ORF8 protein of SARS-CoV-2. Biochem Biophys Res Commun. 2021 Jan 29;538:116-124. doi: 10.1016/j.bbrc.2020.10.045. Epub 2020 Oct 21. PMID: 33685621; PMCID: PMC7577707.
As it stands, the present study does not represent a beneficial addition to the literature. If the authors choose to revise the manuscript for further publication it would be beneficial to consider ways to build on the many existing studies, cite them appropriately, and improve the description of methods (the current "Sequence analysis" paragraph lacks detail).
Author Response
Thank you for the valuable suggestion. We are aware of similar studies in the field, and we will add the peer-reviewed studies to our report (highlighted in red). We have also improved our description of methods as well as make the text more concise (changes highlighted in red). In this brief report, we focus on providing experimental data related to the genotype, which is not demonstrated in related studies. Most studies on orf8 evolution provide information based on sequence analysis, but not from live virus infection. However, it is crucial to understand whether orf8 is being secreted despite the presence of full-length sequences, such as the case in Delta and BA.5. We hope to provide a small piece of experimental evidence to show the effect during virus infection and hope that it can be further elaborated in the future.
Reviewer 2 Report
Comments and Suggestions for Authors
In this brief report, the authors primarily focus on the role of ORF8 deficiency patterns in the dominant SARS-CoV-2 variants, the impact of mutations on ORF8 expression, and the evolutionary trajectory of ORF8-deficient variants such as Alpha, Delta, BA.5.2, and XBB.1.5. The report also highlights the emergence of variants with full-length ORF8, including Omicron BA.1/BA.2 and JN.1.
The manuscript is well-written and provides a thorough discussion, and the current version is suitable for consideration.
Minor comments:
- The paper’s discussion mainly correlation between ORF8 evolution, mutations, and expression levels detected via Western blot is a key strength. However, much of the content in this brief report has already been addressed in another research article (web), with the exception of the Western blot expression levels. (Virological.com: https://virological.org/t/repeated-loss-of-orf8-expression-in-circulating-sars-cov-2-lineages/931 ) It would be helpful to reference this article, as the research topic and data are very similar, apart from the Western blot image.
Author Response
Thank you for reviewing and giving us comments. We are aware of similar publications stating the deletion of orf8 in certain strains. In this brief report, we focus on providing a piece of experimental evidence during live virus infection, while other published studies are based on sequence analysis. We hope that this piece of evidence can facilitate the future discovery of the orf8 function and elucidation of the impact of the recurring changes of orf8. We have added more peer-reviewed references regarding other similar studies to our manuscript (highlighted in red) and we have also modified the text to further improve clarity.
Reviewer 3 Report
Comments and Suggestions for Authors
Lam et al in their manuscript titled “The SARS-CoV-2 orf8 is a unique accessory viral protein in human coronaviruses” present the recurring pattern of orf8 deficiency in dominant SARS-CoV-2 strains. They revealed the substitutions and deletions within SARS-CoV-2 Orf8, identified several high-frequency mutations in in XBB.1.5, Alpha, Delta, and BA.5.2 variants, and experimentally determined that Delta, BA.5.2, and other variants have lost orf8 secretion during infection, thus suggesting that the deficiency may be periodic and could impact SARS-CoV-2 virology. The findings are interesting and deserved further investigation.
In term of the findings, I have no doubt that it might lead to further exploration on the functions and evolution of orf 8, which will enrich the understanding of pathogenesis underlying the SARS-CoV-2 dominant mutants. However, the manuscript needs to be modified significantly.
1. Line 43-44 “Many studies on orf8 protein have been carried out, and it seems to be a multifunctional viral protein involved in various processes, such as impairing host antigen presentation”. This sentence is described ambiguously and should be changed to “orf8 protein has been studies and demonstrated to play a multifunctional functions involved in various processes, such as impairing host antigen presentation”
- Line 47-49 “In our previous work, as well as studies performed by others, the presence of extracellular orf8 protein was confirmed, leading to the fact that orf8 is the first secreted viral protein among human coronaviruses” change to “Previous studies from our group and others confirmed the presence of extracellular orf8 protein, leading to the fact that orf8 is the first secreted viral protein among human coronaviruses”
- Line 53-53 “But whether its role is essential to the virus fitness, or it is beneficial but dispensable, remains a question to be answered.” should changes to “But its role to virus fitness remains to be answered.”
- Line 54-70 This paragraph should be briefly described or outline the method, the results, and its significance. Here the author provided too much detailed results, needing modifying.
- In result sections, passive voice should be used and first person (we) narration should be avoided.
- In the discussion section, the results should not repeated.
Comments on the Quality of English Language
need concise and clear!
Author Response
Thank you for the valuable comments. We have followed your comments and modified the manuscript to make it more concise (changes highlighted in red). We have also added more peer-reviewed references regarding other studies.
Round 2
Reviewer 3 Report
Comments and Suggestions for Authors
in line 111, change "Here we..." to "We..." to avoid the "here" repeat in the line 109.
Author Response
Thank you for the comment. We have amended line 111.